# Breastfeeding Practices among Adolescent Mothers and Associated Factors in Bangladesh (2004–2014)

**DOI:** 10.3390/nu13020557

**Published:** 2021-02-08

**Authors:** Kingsley Emwinyore Agho, Tahmeed Ahmed, Catharine Fleming, Mansi Vijaybhai Dhami, Chundung Asabe Miner, Raphael Torome, Felix Akpojene Ogbo

**Affiliations:** 1School of Health Sciences, Campbelltown Campus, Western Sydney University, Campbelltown, NSW 2560, Australia; Catharine.Fleming@westernsydney.edu.au; 2Translational Health Research Institute (THRI), Campbelltown Campus, Western Sydney University, Sydney, NSW 2560, Australia; 18807282@student.westernsydney.edu.au (M.V.D.); F.Ogbo@westernsydney.edu.au (F.A.O.); 3African Vision Research Institute (AVRI), University of KwaZulu-Natal, Durban 4041, South Africa; 4International Centre for Diarrhoeal Disease Research, Bangladesh 68 Shah Heed Tajuddin Ahmed Ave, Dhaka 1212, Bangladesh; tahmeed@icddrb.org; 5Department of Community Medicine, Faculty of Clinical Sciences, College of Health Sciences, University of Jos, Jos 930003, Nigeria; minerc@unijos.edu.ng; 6Barmera Medical Clinic [Lake Bonney Private Medical Clinic], 24 Hawdon Street, Barmera, SA 5345, Australia; rtorome@barmeramedical.com.au; 7General Practice Unit, Prescot Specialist Medical Centre, Welfare Quarters, Makurdi 972261, Nigeria

**Keywords:** breastfeeding, infants, Bangladesh, morbidity, adolescent mothers, mortality

## Abstract

Optimal breastfeeding practices among mothers have been proven to have health and economic benefits, but evidence on breastfeeding practices among adolescent mothers in Bangladesh is limited. Hence, this study aims to estimate breastfeeding indicators and factors associated with selected feeding practices. The sample included 2554 children aged 0–23 months of adolescent mothers aged 12–19 years from four Bangladesh Demographic and Health Surveys collected between 2004 and 2014. Breastfeeding indicators were estimated using World Health Organization (WHO) indicators. Selected feeding indicators were examined against potential confounding factors using univariate and multivariate analyses. Only 42.2% of adolescent mothers initiated breastfeeding within the first hour of birth, 53% exclusively breastfed their infants, predominant breastfeeding was 17.3%, and 15.7% bottle-fed their children. Parity (2–3 children), older infants, and adolescent mothers who made postnatal check-up after two days were associated with increased exclusive breastfeeding (EBF) rates. Adolescent mothers aged 12–18 years and who watched television were less likely to delay breastfeeding initiation within the first hour of birth. Adolescent mothers who delivered at home (adjusted OR = 2.63, 95% CI:1.86, 3.74) and made postnatal check-up after two days (adjusted OR = 1.67, 95% CI: 1.21, 2.30) were significantly more likely to delay initiation breastfeeding within the first hour of birth. Adolescent mothers living in the Barisal region and who listened to the radio reported increased odds of predominant breastfeeding, and increased odds for bottle-feeding included male infants, infants aged 0–5 months, adolescent mothers who had eight or more antenatal clinic visits, and the highest wealth quintiles. In order for Bangladesh to meet the Sustainable Development Goals (SDGs) 2 and 3 by 2030, breastfeeding promotion programmes should discourage bottle-feeding among adolescent mothers from the richest households and promote early initiation of breastfeeding especially among adolescent mothers who delivered at home and had a late postnatal check-up after delivery.

## 1. Introduction

In many low- and middle-income countries (LMICs), approximately 13 million adolescent girls give birth annually [1,2], and Bangladesh has the highest adolescent pregnancy rate in Asia. An estimated 12% of adolescent Bangladeshi girls fall pregnant before the age of 19 years, and this is largely driven by the high rate of child marriage [3]. This early age of marriage and subsequent pregnancy has huge implications for both the mother and baby, as pregnancy and childbirth-related issues were the leading causes of death in adolescent girls in 2017 [4,5], while babies born to adolescent mothers have an increased risk of dying compared to those born to older mothers [6].

Appropriate breastfeeding practices increase child survival and maternal health. The mechanism, as well as the benefits of how breastfeeding increases child survival, are well documented in the scientific literature [7]. Breastfeeding prevents the introduction of potentially contaminated prelacteal foods, provides newborns with colostrum (rich in a variety of nutrients and immunoglobulins) and protects against diarrheal diseases and respiratory tract infection [7,8,9,10,11,12]—two leading causes of preventable under-five-years deaths in Bangladesh [13]. Numerous research from Bangladesh has identified the determinants of breastfeeding (BF) practices among mothers of all age groups [14,15,16,17,18]. However, specific studies that focus on breastfeeding practices of adolescent mothers are limited. Key factors associated with appropriate breastfeeding practices among Bangladeshi mothers of all age groups included non-prelacteal feeding practices [15], a lack of intimate partner violence and/or child abuse [14], access to mass media, home birthing [16], non-caesarean birthing, and receipt of breastfeeding counselling, antenatal and postnatal care [16,18].

In 2016, the World Health Organisation (WHO) and partners called for an accelerated global action to improve adolescent health, with subsequent impact on infants of adolescent mothers [4,19]. In relation to breastfeeding practices in adolescent mothers, evidence from both high-income countries (such as Australia [20,21,22]) and LMICs (e.g., India [23], Brazil [24] and Nigeria [25]) has indicated that adolescent mothers are less likely to breastfeed their infants compared to older mothers. These studies found that barriers such as socioeconomic disadvantage and a lack of a prenatal attitude towards breastfeeding in Australia, and low socioeconomic status and poor health service contact (e.g., no antenatal care visits) prevented the uptake of optimal breastfeeding behaviours in adolescent mothers. This specific information can help policy decision-makers and health practitioners to implement targeted interventions to increase breastfeeding in this age-group. The research also suggested that breastfeeding support and education for adolescent mothers is needed to increase their breastfeeding participation. However, the utility of this evidence to inform breastfeeding interventions among Bangladeshi adolescent mothers is limited due to varied socio-economic, health and political contexts.

Limited studies have examined breastfeeding and complementary feeding practices among adolescent mothers in Bangladesh. For example, qualitative research that investigated knowledge, attitudes and perceptions towards infant and young child feeding (IYCF) among adolescent mothers in two rural regions in north-west Bangladesh found that knowledge of IYCF among adolescent mothers was generally low [26]. Another study that examined trends and determinants of exclusive breastfeeding (EBF) among adolescent mothers using data from both Health and Demographic Surveillance (HDSS) system areas of the International Centre for Diarrheal Disease Research, Bangladesh (icddr,b) service area (ISA) and government service area (GSA) in rural Bangladesh found that EBF prevalence was 43% on average, with no significant associations noted between the study factors and EBF in the multivariate analyses [27]. Limitations of these studies are that adolescent mothers who lived in urban, semi-urban and other rural areas were excluded, and the study findings may not be generalisable to the wider Bangladeshi population. Therefore, this present study aimed to investigate the prevalence and factors associated with breastfeeding practices (i.e., early initiation of breastfeeding, exclusive breastfeeding, predominant breastfeeding and bottle feeding) among adolescent mothers in Bangladesh using nationally representative data, to identify a specific population to provide targeted breastfeeding education and support services.

## 2. Materials and Methods

We used the combined datasets for the years 2004, 2007, 2011 and 2014 from the Bangladesh Demographic and Health Survey (BDHS). A detailed description of the survey methodology, sampling procedure, and questionnaires used for data collection is provided elsewhere [28]. BDHS datasets were collected by Population Research and Training (NIPORT) of the Ministry of Health and Family Welfare. The survey was implemented by Mitra and Associates, a Bangladeshi research firm located in Dhaka. ORC Macro of Calverton, Maryland, provided technical assistance to the project as part of its international Demographic and Health Surveys program. The BDHS covered the entire population residing in private unit dwellings in the country, although the 2004 and 2007 BDHS was administratively divided into six divisions while the 2011 and 2014 BDHS was divided into seven divisions with Rangpur included. The BDHS collects socio-demographic, maternal and child health information from a nation-wide representative sample of households, using a three-stage sampling strategy. 

A weighted total sample of 2554 adolescent mothers aged 12–19 years were included in the final analysis (2004: n = 614; 2007: n = 521; 2011: n = 693; and 2014: n = 726). The average response rate for the four surveys was 98%. For the purpose of this study, we restricted our analysis to the last-born child under 24 months, alive and living with the respondent. A weighted total of 2554 children born to adolescent mothers were assessed for the World Health Organization (WHO) defined breastfeeding indicators.

### 2.1. Outcome and Confounding Factors

In this study, the selected BF variables examined were early initiation of breastfeeding within one hour of birth (EIBF), exclusive breastfeeding (EBF), predominant breastfeeding (PBF) and bottle-feeding (BotBF) because early initiation of breastfeeding and exclusive breastfeeding plays an important role in reducing infant mortality and morbidity [29] while predominant breastfeeding and bottle-feeding are risk factors for diarrhoeal and respiratory illness [30]. The selected breastfeeding indicators that were assessed based on the WHO definitions for assessing IYCF practices [31] were as follows:EBF was defined as the proportion of infants 0–5 months of age who received breast milk only and no other solids or liquids except for vitamins, minerals, medicines or oral rehydration solution;PBF was defined as the proportion of infants 0–5 months of age who received breast milk and water, water-based liquids such as sugar water and juices but not infant formula or milk;EIBF was defined as the proportion of children aged 0–23 months who were put to the breast within one hour of birth and,BotBF was defined as the proportion of children aged 0–23 months who received any food or liquid including non-human milk and formula, any liquid (including breast milk) or semi-solid food from a bottle with nipple/teat.

Confounding factors included in the analysis were guided by previous studies [32,33,34] and were classified into four levels (see Appendix A for categorisation details): individual, exposure to media, household and community level. Individual-level factors related directly to the ‘mother-infant dyad’ described by a previous group [35]. The attributes of the infant included age, the number of children and sex of the baby, while the characteristics of the adolescent mother included mother’s age, working status, husband’s education and occupation, religion, education level, maternal body mass index (BMI) and marital status. Factors related to delivery included antenatal visits, combined place and mode of delivery, type of delivery assistance; birth order and postnatal contact and were classified under attributes of the mother-infant dyad.

Household wealth index and the women’s role in household decisions were included as household-level variables. The household wealth index was constructed using methods recommended by the World Bank Poverty Network and United Nations International Children’s Emergency Fund (UNICEF) and described by Filmer and Pritchett [36] and UNICEF, 2005 [37]. The principal components statistical procedure was used to determine the weights for the wealth index based on information collected about 22 household assets and facilities in order to estimate the household wealth index factor score. The household wealth index factor score (hv271) was constructed by the BDHS for each year of the survey. The household wealth index factor score for the datasets was pooled and was divided into quintiles. We then created a new variable (called the household wealth index) for the quintiles and the bottom 20% of households was arbitrarily referred to as the poorest households, the next 20% as the poorer households and the top 20% as the richest households as used in past studies [23,38] (Appendix A). 

In BDHS surveys, women’s decision-making autonomy was assessed by information on women’s participation in three different types of decisions: the person who usually decides on the respondent’s health care, the person who usually decides on making large household purchases and the person who usually decides on visits to family or relatives. We categorised each of the three different types of decisions as category 1 for ‘respondent alone’ and ‘respondent and husband/partner’ and category 0 for ‘husband/partner alone’, ‘respondent and another person’ and someone else. We then created a composite score ranging from 0 to 3 points which was then categorised into three, namely no decision (0 score), some decisions (1–2 scores) and all decisions (3 scores). Exposure to media variables included newspapers, radio and television and the community level factors included the residential region of residence and geographical region.

### 2.2. Statistical Analysis

First, the selected BF indicators were dichotomized as category 1 for EBF, BotBF and PBF, and category 0 for not EBF, BotBF and PBF while, delayed initiation of breastfeeding was categorised as 1 and EIBF as 0. 

Second, the sampling weights were different by survey and hence we re-normalised the sampling weights for each year of the survey to add up to 1 by computing the total sum of weights for each survey round and dividing each year of survey sampling weights with the total sum of weights. This was followed by frequency tabulations to describe the characteristics of the study population and descriptive analysis for estimating 95% confidence intervals (CI) around prevalence estimates for all the breastfeeding indicators and linear interpolation was used to obtain the median duration of any breastfeeding and EBF in the month.

Third, for the selected BF indicator variables, univariate logistic regression analysis was used to present as unadjusted OR (95% CI) for each confounding variable while multivariate logistic regression was used to identify factors associated with the selected breastfeeding indicators.

In the multivariate logistic regression analysis, four-stage modelling was employed. In the first stage, the individual level factors were entered into the first stage model (model 1). We conducted a manually executed elimination method to determine factors associated with each selected breastfeeding indicator (*p* < 0.05). The significant factors in Model 1 were then added to the exposure to media factors in the second stage model (model 2); this was then followed by a manually executed elimination procedure. The significant factors for combined Models 1 and 2 were added to the household factors in the third model (Model 3). We used a similar approach for community-level factors in the fourth stage (Model 4), respectively. Factors associated were presented as adjusted OR (95% CI) for the variables retained in the final modelling step. These analyses were performed using Stata ‘svy’ commands that allow for adjustments of cluster sampling design and weights. STATA V.14.1 (STATA Corporation, College Station, TX, USA, 2015) was employed for all the study analyses.

## 3. Results

### 3.1. Characteristics of the Sample

Table 1 lists the individual, household and community-level characteristics of the children included in this analysis. Nearly one-tenth (9.2%) of adolescent mothers had worked in the past 12 months prior to the survey, out of which, more than a third (34.9%) had a secondary or higher level of education. Approximately one in four adolescent mothers (39.5%) were from the poorer or poorest household wealth quintiles, and the majority (80%) were from rural areas. Of the respondents, 86% never read newspapers, 76% never listened to the radio, and 41% never watched television. Low BMI (<18.5 kg/m^2^) was observed in 30%, one-third (29.9%) had no antenatal clinic visit during pregnancy, 75.6% were home deliveries, 21.5% had health professionals available at delivery, and 12.1% were delivered by caesarean section. The frequency distribution of the individual, exposure to media, household and community level characteristics of children 0–23 months of age by year of the survey are presented (see, Appendix A).

### 3.2. Breastfeeding and Infant Feeding Indicators

As shown in Table 2, of the total 2554 children aged 0–23 months, the ever-breast-fed rate was universal (99.7%), and 99.7% were currently breastfed during the previous day or night. The breastfeeding continuation in the first year was 96.2% and in the second year 91.4%. EIBF was observed in only 42.2% and the exclusive breastfeeding rate in children less than 6 months of age was 53.0%. BotBF in the last 24 h was noted in 15.7%, and only 64.9% were introduced to complementary feeds early. The median duration of exclusive breastfeeding 0–6 months was 3.6 months and for any breastfeeding was 23 months among the children aged 0–23 months. Key breastfeeding indicators among children 0–23 months of age examined as the outcome variables by year of the survey are presented (see, Appendix A).

Figure 1 shows the proportion of children born to adolescent mothers by breastfeeding status according to age from 2004 to 2014. As illustrated in Figure 1, the EBF at birth was above 87% and declined rapidly to 43% by 4 months of age. At birth, infants’ proportion of breastfed plus water was 2.7% and increased to about 16% at 2 months. Similarly, breastfed plus other milk was 7.2% at birth and increased to 16.2% by 2 months of age (see, Appendix A for details).

### 3.3. Determinants of Selected Feeding Indicators

Unadjusted and adjusted odds ratios (OR) were calculated to estimate the strength of association between independent variables and four infant feeding outcomes: (1) delayed initiation of breastfeeding; (2) BotBF, (3) EBF and, (4) PBF. As shown in Table 3 and Table 4.

#### 3.3.1. Factors Associated with Delayed Initiation of Breastfeeding and Bottle-Feeding

As shown in Table 3, adolescent mothers who delivered at home (adjusted OR = 2.60, 95%CI: 1.84, 3.69; *p* < 0.001), had no ANC (antenatal care) visit during pregnancy (adjusted OR = 1.83, 95%CI: 1.13, 2.97; *p* = 0.014) and had a postnatal check-up two days after delivery (adjusted OR = 1.70, 95%CI: 1.24, 2.35; *p*= 0.001) were more likely to delay initiation of breastfeeding. Adolescent mothers aged 18–19 years were less likely to delay initiation of breastfeeding (adjusted OR= 0.78, 95%CI: 0.65, 0.94; *p* = 0.008 for adolescent mothers aged 18–19 years). Delayed initiation of breastfeeding was lower among adolescent mothers who watched TV (adjusted OR = 0.76, 95%CI: 0.63, 0.93; *p* = 0.006) and lived in the Rangpur region (adjusted OR = 0.54, 95%CI: 0.36, 0.82; *p* = 0.004).

Adolescent mothers who had higher secondary or more educational status (adjusted OR = 0.57, 95%CI: 0.32,0.99; *p* = 0.050) were less likely to bottle feed their children (Table 3). Being a female infant (adjusted OR = 0.70 95%CI: 0.51, 0.96; *p* = 0.026) and children aged 18–23 months of age (adjusted OR = 0.61 95%CI: 0.38, 0.98; *p* = 0.043) were less likely to bottle-feed. Adolescent mothers who did not have any antenatal care visit during pregnancy reported a decreased likelihood of bottle feeding (adjusted OR = 0.38, 95%CI: 0.18, 0.80 *p* = 0.011) compared to adolescent mothers who had eight or more antenatal care visits during pregnancy. Compared to adolescent mothers from the richest wealth quintiles, adolescent mothers from middle (adjusted OR = 0.57 95%CI: 0.36, 0.90; *p* = 0.015), poor (adjusted OR = 0.38 95%CI: 0.22, 0.64; *p* < 0.001) and poorest (adjusted OR = 0.25 95%CI: 0.14, 0.44; *p* < 0.001) wealth quintiles had a decreased likelihood of bottle feeding (see columns 14–17 in Table 3 for details) 

#### 3.3.2. Factors Associated with Exclusive Breastfeeding and Predominant Breastfeeding

As expected, EBF was more likely in younger children (Table 4) because the child’s age increases as EBF decreases. Adolescent mothers who had more than one child reported higher EBF and those adolescent mothers who had postnatal care after 2 days had a higher likelihood of EBF. The infants from adolescent mothers who reside in Chittagong, Khulna, Sylhet and Rangpur significantly had a positive impact on EBF. 

The child’s age increases as PBF increases. Adolescent mothers who reside in Chittagong, Dhaka, Rajshahi and Rangpur significantly reported lower PBF and adolescent mothers who listened to the radio had a greater likelihood of PBF. 

## 4. Discussion

Our results indicated that EBF and EIBF rates among adolescent mothers were sub-optimal because less than half of adolescent mothers had EIBF, and that only half of the adolescent mothers EBF their infants aged less than 6 months and need further improvement in order to gain the full benefits of breastfeeding including health and nutritional status of their children. Multivariate analyses showed that child age in months, parity, exposure to media (radio), postnatal check-up after 2 days and geographical region were associated with EBF and PBF. EIBF was greater among adolescent mothers aged 18–19 years who had exposure to media (television). Infants delivered at home and had no ANC and had a postnatal check-up after two days after delivery reported a decreased likelihood of EIBF. Children from adolescent mothers who had secondary or more educational status, female infants, children aged 18–23 months, children from adolescent mothers who had no ANC and children from the middle, poor and poorest wealth quintiles had a decreased likelihood of bottle feeding. 

Our study found that adolescent mothers aged <18 years reported a decreased likelihood of EIBF compared to those aged 18–19 years. This finding was consistent with a cross-sectional study conducted in Nigeria which found that EIBF among adolescent mothers aged 12–18 years reduced by the crude odds of 36% compared with young women (20–24.9 years) [38]. Younger mothers lack experience, education, attitude and the necessary social support required for the appropriate breastfeeding practices [26]. We identified the relationship between exposure to media (television) and EIBF. A study conducted in 30 countries in Sub-Saharan Africa revealed that women who watched television had a higher likelihood of making a beneficial decision about breastfeeding [39]. Additionally, the association could also be attributed to the fact that adolescent mothers had a higher likelihood of being exposed to media campaigns and public health messaging regarding appropriate IYCF practices [26] because in South Asia, a higher economic status strongly correlates with access to media [40].

It was observed that adolescent mothers who had no ANC visit during pregnancy reported delayed initiation of breastfeeding. This finding supported a cross-sectional study conducted in Bangladesh [18] and India [23] that no ANC visit during pregnancy was less likely to EIBF but contradicts a cross-sectional study that examined trends and factors associated with breastfeeding practices among adolescent and younger mothers in Nigeria that found adolescent mothers who had ANC were less likely to delay breastfeeding initiation [38]. A past study conducted in Bangladesh indicated that among women that had ANC, about half of them were accompanied by their partners/husbands to the health facility [40]. Partners/husbands play a significant role in improving breastfeeding, including EIBF [41]. BF information to partners should be timely at antenatal and postnatal periods, and partner-to-partner BF support groups using social media such as Facebook are also needed to improve BF including EIBF among adolescent mothers in Bangladesh.

Delayed breastfeeding initiation is more prevalent among adolescent mothers who delivered at home and those who had postnatal care visit 2 days after delivery. These findings are consistent with a cross-sectional study conducted in Nigeria, and secondary analysis of the WHO Global Survey conducted in 24 countries which found that adolescent mothers who utilise a health service (delivered at the health facility and early postnatal care) were more likely to initiate breastfeeding within one hour of birth and late or absent postnatal care affected EIBF [42]. Although the finding that adolescent mothers who delivered at the health facility were more likely to initiate breastfeeding within one hour of birth seems counterintuitive because the previous study in Bangladesh revealed that mothers aged 18–49 years were more likely to initiate breastfeeding within one hour of birth if they give birth at home rather than in a health facility [43]. However, community-based breastfeeding counselling intervention including health education messages similar to those received through the Baby-Friendly Hospital Initiative (BFHI) may be needed to increase the likelihood of EIBF among those adolescent mothers who delivered at home.

Adolescent mothers who lived in the Khulna and Rangpur divisions were more likely to initiate early breastfeeding compared with adolescent mothers who lived in the Barisal division. The finding that the Rangpur division was more likely to initiate EIBF supported a previous study on the prevalence and determinants of EIBF in Bangladesh conducted in 2019 [43], although our findings could be attributed to the differences in the proportion of adolescent women who initiated breastfeeding early. However, further research is needed to explore the differences in region-specific cultural practices that could be associated with EIBF to improve the effectiveness of the EIBF program among adolescent mothers in the country.

Adolescent mothers who had higher education were less likely to bottle-feed their children than those with no schooling. This finding was supported by a community-based cross-sectional study conducted among economically disadvantaged mothers in Karachi, Pakistan which found that illiterate and less educated mothers were significantly more likely to bottle-feed their children than more educated mothers [44]. The increased likelihood of bottle-feeding among adolescent mothers with no schooling may be due to limited economic resources and higher maternal education increases opportunities to make informed child health-related decisions. These decisions include the uptake and practice of appropriate IYCF, which can improve a mother’s attitude towards seeking appropriate child health support for appropriate IYCF [45]. Additionally, our study found that female infants were less likely to be bottle-fed than male infants, and adolescent mothers from the poor and poorest wealth quintiles were less likely to bottle-feed their children compared to those from the richest wealth quantiles. These findings were inconsistent with a cross-sectional study conducted on bottle-feeding in Indonesia which found that the richest families reported an increased likelihood of bottle feeding and female children were less likely to be bottle-fed than their counterparts [46]. The significantly lower odds of bottle feeding practice among poor mothers reported in this study may be attributed to the inability to finance or have access to infant formula or costly complementary substitutes to breast milk, which may result in poor adolescent mothers depending solely on breastfeeding [47].

Older infants aged 18–23 months were less likely to receive BotBF, but the main effect of children’s ages across the categories did not differ statistically (Adjusted Wald χ^2^ = 2.22, *p*= 0.085). This finding contradicts previous studies that reported that the increased age of an infant was associated with bottle-feeding [46,48]. There is also a possibility that this finding could be affected by the number of children bottle-fed in different age categories with about one-third of children aged 0–17 months being bottle-fed compared to 16.2% of children aged 18–23 months. We found that adolescent mothers who had no antenatal clinic (ANC) visits during pregnancy were less likely to bottle-feed their children compared to those adolescent mothers who had eight or more ANC visits during pregnancy. The finding was supported by a cross-sectional study conducted in Pakistan which reported that the increased likelihood of bottle-feeding among adolescent mothers who had adequate (four or more) ANC visits during pregnancy [49]. However, it was unexpected to find that mothers who had adequate ANC visits during pregnancy were more likely to bottle-feed their children than those who had no visits because of the breastfeeding counselling and health education messages received through the Baby-Friendly Hospital Initiative (BFHI). Hence, there is a need for community-based health education interventions (such as father/partner involvement in breastfeeding [50]) and nutritionally appropriate information sessions for new adolescent mothers to have better opportunities to make informed decisions about their child’s health.

In our study, adolescent mothers who had two or more children, younger infants, living in Chittagong, Khulna, Sylhet and Rangpur divisions had higher odds of EBF. These findings are consistent with a past cross-sectional study conducted in Bangladesh, and 13 Economic Community of West African States (ECOWAS) countries which found that higher birth order was associated with increased exclusive breastfeeding rates [16,32,51]. The finding that adolescent mothers who had older infants reported lower odds of EBF could be attributed to the duration of maternity leave because infant adolescent mothers may be forced to return to full-time work with a shorter breastfeeding span given Bangladesh’s recent impressive economic growth and changes in work practices, such as part-time work which has been reported to increase breastfeeding span [52]. These findings may be related to inexperience regarding breastfeeding among first time adolescent mothers including inadequate use of maternal health services and a lack of effective breastfeeding intervention programs among adolescent mothers who worked at a factory, with previous research finding an association between a mother’s age and inadequate use of maternal health services [53]. 

PBF was less likely in younger infants than older infants. This finding was consistent with a cross-sectional study conducted in Malawi, which revealed that the odds of PBF was higher in older infants than younger infants under 6 months of age [54] and PBF (BF plus water) should be discouraged at 6 months because it is a risk factor for diarrhoeal and respiratory illness in infants while EIBF and EBF should be encouraged because they play a vital role in protecting infants against diarrhoeal diseases and reduce mortality [55].EBF increased among adolescent mothers who had a postnatal check-up after 2 days and a prospective cohort study on the effect of timing of the first postnatal care home visit on neonatal mortality conducted in Bangladesh revealed that neonatal mortality reduced by about 21% among women who had a postnatal check-up after 2 days after delivery compared to those with no postnatal check-up [56]. 

Our study found that PBF increased among adolescent mothers who listened to the radio more than those who did not listen to the radio. This finding supported a study conducted in Bangladesh which suggested the need to reinforce breastfeeding messages through various mass media and backed these massages during ANC visits [57]. The survey conducted by Faruque et al. [57] further suggested the need to closely monitor those breastfeeding messages to ensure that they are exact and should reach adolescent mothers with low literacy levels and limited exposure to electronic and print media. Adolescent mothers living in Chittagong, Dhaka, Rajshahi and Rangpur divisions reported increased odds of predominant breastfeeding and predominant breastfeeding was less likely in younger infants, and a past study revealed that Chittagong, Dhaka and Rajshahi reported a lower prevalence of PBF than Sylhet [51].

This study has several strengths and limitations. First, this was the first study to examine breastfeeding practices among adolescent mothers and associated factors in Bangladesh using nation-wide population-based surveys because appropriate breastfeeding practices contributed to two Sustainable Development Goals (SDGs)—SDG 2 and SDG 3—which are aimed at improving nutrition and increasing maternal and child health. Second, the study has great statistical power because four datasets (2004, 2007, 2011, 2014) were combined to create a large sample size and to detect statistical differences. Third, our analysis was restricted to children alive and living with the respondent to reduce adolescent mothers recall bias and are less likely to be influenced by selection bias due to an extremely high response rate, and the data collection was done over a validated standardised questionnaire. However, this study also has some limitations. First, the cross-sectional study design limits causal inference and recall bias may have influenced our findings because self-reported data were collected and analysed. Second, it is a possibility that there could have been some misclassification bias and that may have led to an overestimation or underestimation of the relevant breastfeeding indicators. Third, we did not account for all the confounding factors (e.g., cultural beliefs, birth weight, influences of food safety measures) that may have impacted factors associated with breastfeeding practices among adolescent mothers are not included in the analysis. Fourth, bottle-feeding was not collected in the 2007 BDHS. Fifth, it is difficult to establish a clear temporal association between the factors associated and the breastfeeding practices due to the cross-sectional data of the study. Lastly, some maternal and infant health factors (such as gestational age, Apgar score, birth weight and maternal gynecological history, e.g., miscarriage) may be important variables for breastfeeding outcomes but were not examined in this study due to data availability.

## 5. Conclusions

The present study suggests considerable variations in the prevalence of the key breastfeeding indicators among adolescent mothers in Bangladesh. Factors associated with the breastfeeding indicators also varied across the individual, community, and national level. At the individual level, BF information to adolescent mothers and their partners should be early and consistent at hospitals, birthing units, and from community midwives and health visitors from the Upazila. At the household level, increasing education for adolescent mothers on the best BF practices, appropriate BF education to partners and mothers-in-law of adolescent mothers about breastfeeding, and improving health service use for both the adolescent mother and her baby should be encouraged. At the community level, improving geography-specific access to breastfeeding messages will improve breastfeeding practices of adolescent mothers.

## Figures and Tables

**Figure 1 nutrients-13-00557-f001:**
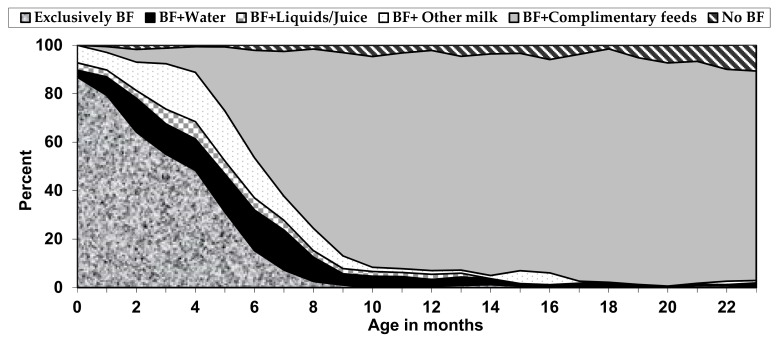
Distribution of children born to adolescent mothers by breastfeeding (BF) status, according to age in Bangladesh 2004–2014 (n = 2554).

**Table 1 nutrients-13-00557-t001:** Individual, exposure to media, household and community level characteristics of children 0–23 months of age, Bangladesh 2004–2014 (n = 2554).

Characteristic	n	%
***Individual-level factors***		
**Mother’s religion (n = 2553)**		
Islam	2374	93.0
Others ^$^	179	7.0
**Maternal working status**		
Non-working	2320	90.8
Working (past 12 months)s	234	9.2
**Maternal education**		
No education	268	10.5
Primary	811	31.8
Secondary and higher	1475	57.8
**Partner’s education**		
No education	596	23.3
Primary	1064	41.7
Secondary and higher	892	34.9
**Husband’s occupation (n = 2552)**		
Non-agricultural	1582	61.9
Agricultural	667	26.1
Not working	305	12.0
**Mother’s age**		
12–18 years	1062	41.6
18–19 years	1492	58.4
**Marital status**		
Currently married	2527	99.0
Formerly married ^	26	1.0
**Birth order**		
First-born	2083	81.6
2nd–4th	471	18.4
**Preceding birth interval**		
No previous birth	2083	81.6
Yes	467	18.3
**Sex of baby**		
Male	1303	51.0
Female	1251	49.0
**Number of living children**		
1	2157	84.5
2–4	396	15.5
**Age of child (in months)**		
0–5	745	29.2
6–11	717	28.1
12–17	628	24.6
18–23	464	18.2
**Combined mode and place of delivery (n = 2548)**		
Caesarean	299	12.1
Vaginal	289	11.9
Home	1955	75.7
**Type of delivery assistance (n = 2543)**		
Health professional	549	21.2
Non-health professional	2005	78.4
**Antenatal Clinic (ANC) visits**		
8+	104	4.1
4–7	508	19.9
1–3	1179	46.2
None	763	29.9
**Postnatal check-up**		
0–2 days	631	24.7
After 2 days	295	11.6
No postnatal check-up	1628	63.8
**Mother’s BMI (n = 2521)**		
<18.5	754	29.5
18.5–24.9	1675	65.6
25+	92	3.6
Exposure to Media		
**Mother reading newspapers (n = 2552)**		
Not at all	2196	86.0
Yes ^#^	355	13.9
**Mother listening to radio**		
Not at all	1950	76.3
Yes ^#^	604	23.7
**Mother watching television (n = 2552)**		
Not at all	1056	41.4
Yes ^#^	1496	58.6
***Household-level factors***		
**Household wealth Index**		
Richest	497	19.5
Richer	542	21.2
Middle	507	19.9
Poorer	503	19.7
Poorest	505	19.8
**Decision-making (Autonomy)**		
No Decision (0 scores)	1061	41,5
Some Decisions (1–2 scores)	801	31.4
All Decisions (3 scores)	693	27.1
***Community-level factors***		
**Residence**		
Urban	516	20.2
Rural	2038	79.8
**Geographical Region without Rangpur**		
Barisal	145	6.1
Chittagong	534	22.5
Dhaka	791	33.3
Khulna	266	11.2
Rajshahi	494	20.8
Sylhet	145	6.1
**Geographical Region with Rangpur**		
Barisal	145	5.7
Chittagong	534	20.9
Dhaka	791	31.0
Khulna	266	10.4
Rajshahi	494	19.4
Sylhet	145	5.7
Rangpur	178	7.0

^ divorced/separated/widow, ^$^ Hinduism/Buddhism/Christianity/others; ^#^ less than once a week/at least once a week/almost every day. Total count 2554 unless otherwise given in brackets.

**Table 2 nutrients-13-00557-t002:** Breastfeeding indicators among children 0–23 months of age, Bangladesh 2004–2014 (n = 2554).

Breastfeeding Indicators	N *	n *	Rate (95% CI)
**Early initiation of breastfeeding rate ^a^**			
Yes	2554	1078	42.2 (39.7,44.8)
**Ever breastfed rate ^a^**			
Yes	2554	2546	99.7 (99.3, 99.9)
**Bottle-feeding rate ^a^^**			
Yes	2032	319	15.7 (13.7, 17.9)
**Current breastfeeding rate ^a^**			
Yes	2554	2546	99.7 (99.3, 99.9)
**Age-appropriate breastfeeding**			
Yes	2554	1938	75.9 (73.8,77.8)
**Exclusive breastfeeding rate ^b^**			
Yes	745	395	53.0 (48.6, 57.3)
**Predominant breastfeeding rate ^b^**			
Yes	745	129	17.3 (14.4,20.6)
**Continued breastfeeding rate (1 year) ^c^**			
Yes	413	397	96.2 (93.1, 97.9)
**Continued breastfeeding rate (2 years) ^d^**			
Yes	284	260	91.4 (86.6, 94.6)
**Early introduction of complementary feeding rate ^e^**			
Yes	361	234	64.9 (59.0, 70.4)
Median duration of any breastfeeding a (months)			23.0
Median duration of exclusive breastfeeding a (months)			3.6

N * = weighted population; n * = weighted sample; ^ bottle feeding not correct in 2007 BDHS; ^a^—children under 24 months; ^b^—infants below 6 months; ^c^—children 12–15 months; ^d^—children 20–23 months; ^e^—infants 6–8 months.

**Table 3 nutrients-13-00557-t003:** Survey logistic modelling of a child bottle-fed and delayed initiation of BF—unadjusted and adjusted Odds Ratios, Bangladesh 2004–2014.

Characteristic	Delayed Initiation of BF								BotBF							
	Unadjusted				Adjusted				Unadjusted				Adjusted			
	OR	[95% CI]		*p*	OR	[95% CI]		*p*	OR	[95% CI]		*p*	OR	[95% CI]		*p*
***Individual-level factors***																
**Mother’s religion**																
Others ^$^ (Islam, OR = 1)	0.81	0.57	1.16	0.247					0.62	0.32	1.18	0.144				
**Maternal working status**																
Working (Non-working, OR = 1)	0.87	0.64	1.17	0.359					0.90	0.52	1.57	0.703				
**Marital status**																
Formerly married ^^^ (Currently married, OR = 1)	0.88	0.39	1.99	0.760					1.01	0.20	4.97	0.991				
**Maternal education**																
Primary (No education, OR = 1)	0.80	0.57	1.12	0.197					0.92	0.54	1.60	0.776	0.65	0.36	1.15	0.139
Secondary and above	0.82	0.59	1.13	0.214					1.25	0.73	2.12	0.417	0.57	0.32	1.00	0.049
**Partner’s education**																
Primary (No education, OR = 1)	0.97	0.71	1.32	0.826					0.86	0.48	1.53	0.606				
Secondary and above	0.72	0.55	0.94	0.015					1.30	0.76	2.21	0.334				
**Husband’s occupation**																
Agricultural (Non-agricultural, OR = 1)	0.99	0.80	1.23	0.939					0.72	0.43	1.20	0.212				
Not working	0.77	0.58	1.03	0.080					0.96	0.63	1.46	0.851				
**Mother’s age**																
18–19 years (12–18 years, OR = 1)	0.81	0.68	0.98	0.028	0.79	0.66	0.95	0.014	1.00	0.73	1.37	0.995				
**Birth order**																
2nd–4th (First-born, OR = 1)	0.99	0.77	1.27	0.948					0.66	0.43	1.01	0.054				
**Preceding birth interval**																
Yes (No previous birth, OR = 1)	0.98	0.76	1.26	0.862					0.60	0.39	0.94	0.024				
**Sex of baby**																
Female (Male, OR = 1)	1.09	0.91	1.32	0.354					0.70	0.51	0.96	0.026	0.70	0.51	0.96	0.026
**Number of children**																
2–4 (I child, OR = 1)	1.01	0.77	1.33	0.937					0.56	0.35	0.89	0.014				
**Age of child (months)**																
6–11 (0–5 months, OR = 1)	0.93	0.69	1.24	0.606					1.31	0.83	2.05	0.244	1.30	0.85	1.99	0.233
12–17	1.07	0.84	1.37	0.560					0.85	0.58	1.27	0.432	0.88	0.58	1.33	0.545
18–23	0.95	0.72	1.26	0.730					0.63	0.40	0.99	0.047	0.61	0.38	0.98	0.043
**Combined Mode and Place of delivery**																
Vaginal (Caesarean, OR = 1)	0.83	0.64	1.08	0.163	1.07	0.79	1.45	0.683	1.29	0.83	2.00	0.257				
Home	1.93	1.42	2.63	<0.001	2.60	1.86	3.69	1.93	1.31	2.85	0.001					
**Type of delivery assistance**																
Non-health professional (Health professional, OR = 1)	0.79	0.64	0.98	0.029					0.62	0.45	0.86	0.004				
**Antenatal Clinic visits**																
4–7 (8+, OR = 1)	1.30	0.82	2.06	0.259	1.52	0.96	2.41	0.076	0.89	0.41	1.94	0.763	0.99	0.46	2.11	0.978
1–3	1.18	0.75	1.85	0.482	1.34	0.85	2.12	0.204	0.60	0.30	1.20	0.150	0.72	0.36	1.42	0.341
None	1.60	1.00	2.57	0.050	1.83	1.13	2.97	0.014	0.28	0.13	0.58	0.001	0.38	0.18	0.80	0.011
**Postnatal check-up**																
After 2 days (0–2 days, OR = 1)	1.61	1.17	2.20	0.003	1.70	1.24	2.35	0.001	0.91	0.56	1.48	0.707				
No postnatal check-up	1.12	0.89	1.40	0.337	1.26	0.96	1.64	0.091	0.62	0.44	0.89	0.008				
**Mother’s BMI**																
18.5–24.9 (<18.5, OR = 1)	1.03	0.65	1.65	0.885					0.75	0.41	1.36	0.336				
25+	1.16	0.71	1.91	0.548					0.53	0.28	1.00	0.052				
**Exposure to Media**																
**Mothers reading Newspapers**																
Yes ^#^ (Not at all, OR = 1)	1.07	0.80	1.43	0.661					1.60	1.10	2.34	0.014				
**Mothers listening to the radio**																
Yes ^#^ (Not at all, OR = 1)	1.28	1.02	1.60	0.033					1.30	0.92	1.83	0.137				
**Mothers watching TV**																
Yes ^#^ (Not at all, OR = 1)	0.81	0.67	0.98	0.028	0.76	0.63	0.94	0.006	1.72	1.15	2.57	0.009				
***Household-level factors***																
**Household Wealth Index**																
Richer (Richest, OR = 1)	1.14	0.86	1.50	0.359					0.68	0.43	1.08	0.104	0.75	0.47	1.20	0.225
Middle	1.07	0.78	1.45	0.685					0.48	0.31	0.75	0.001	0.57	0.36	0.90	0.015
Poorer	1.05	0.80	1.38	0.729					0.32	0.19	0.53	<0.001	0.38	0.22	0.64	<0.001
Poorest	1.00	0.72	1.37	0.976					0.22	0.13	0.39	<0.001	0.25	0.14	0.44	<0.001
**Decisions women have a final say**																
Some Decisions (No Decision, OR = 1)	1.06	0.85	1.32	0.581					0.97	0.69	1.36	0.877				
All Decisions	0.82	0.65	1.03	0.081				0.82	0.54	1.21	0.313					
**Residence**																
Rural (Urban, OR = 1)	0.96	0.77	1.19	0.700					0.63	0.46	0.86	0.003				
**Geographical Region without Rangpur**																
Chittagong (Barisal, OR = 1)	1.03	0.71	1.48	0.890					0.84	0.46	1.53	0.564				
Dhaka	0.91	0.64	1.29	0.584					2.21	1.27	3.86	0.005				
Khulna	0.68	0.46	0.98	0.041					0.72	0.38	1.35	0.308				
Rajshahi	0.83	0.59	1.19	0.310					1.29	0.71	2.34	0.401				
Sylhet	0.77	0.53	1.11	0.166					1.21	0.64	2.28	0.560				
**Geographical Region with Rangpur**																
Chittagong (Barisal, OR = 1)	1.03	0.71	1.48	0.890	1.12	0.77	1.63	0.567	0.84	0.46	1.53	0.564				
Dhaka	0.91	0.64	1.29	0.584	0.98	0.69	1.40	0.913	2.21	1.27	3.86	0.005				
Khulna	0.68	0.46	0.98	0.041	0.73	0.51	1.08	0.120	0.72	0.38	1.35	0.308				
Rajshahi	0.83	0.59	1.19	0.310	0.92	0.64	1.12	0.656	1.29	0.71	2.34	0.401				
Sylhet	0.77	0.53	1.12	0.168	0.79	0.54	1.16	0.211	1.21	0.64	2.28	0.559				
Rangpur	0.49	0.33	0.74	0.001	0.54	0.36	0.82	0.004	0.72	0.31	1.71	0.461				

^ divorced/separated/widow, ^$^ Hinduism/Buddhism/Christianity/others; ^#^ less than once a week/at least once a week/almost every day.

**Table 4 nutrients-13-00557-t004:** Survey logistics modelling of a child being breastfed and predominantly breastfed (PBF)—unadjusted and adjusted Odds Ratios, Bangladesh 2004–2014.

Characteristic	EBF								PBF							
	Unadjusted				Adjusted				Unadjusted				Adjusted			
	OR	[95% CI]		*p*	OR	[95% CI]		*p*	OR	[95% CI]		*p*	OR	[95% CI]		*p*
***Individual-level factors***																
**Mother’s religion**																
Others ^$^ (Islam, OR = 1)	1.74	0.84	3.58	0.133					0.80	0.23	2.86	0.735				
**Maternal working status**																
Working (Non-working, OR = 1)	0.64	0.30	1.39	0.262					2.53	1.02	6.29	0.045				
**Marital status**																
Formerly married ^^^ (Currently married, OR = 1)	*	*	*	*					*	*	*	*				
**Maternal education**																
Primary (No education, OR = 1)	0.70	0.37	1.32	0.263					1.07	0.46	2.48	0.876				
Secondary and above	0.76	0.44	1.31	0.326					1.74	0.80	3.78	0.160				
**Partner’s education**																
Primary (No education, OR = 1)	1.24	0.78	1.97	0.360					0.87	0.50	1.51	0.615				
Secondary and above	1.48	0.92	2.38	0.102					0.84	0.47	1.50	0.566				
**Husband’s occupation**																
Agricultural (Non-agricultural, OR = 1)	0.98	0.64	1.50	0.921					1.37	0.80	2.35	0.252				
Not working	1.13	0.64	2.00	0.669					1.09	0.58	2.05	0.791				
**Mother’s age**																
18–19 years (12–18 years, OR = 1)	1.30	0.89	1.88	0.170					1.07	0.70	1.64	0.758				
**Birth order**																
2nd–4th (First-born, OR = 1)	1.38	0.89	2.15	0.149					0.82	0.49	1.36	0.439				
**Preceding birth interval**																
Yes (No previous birth, OR = 1)	1.38	0.89	2.15	0.149					0.82	0.49	1.36	0.439				
**Sex of baby**																
Female (Male, OR = 1)	1.23	0.85	1.79	0.278					0.69	0.45	1.05	0.082				
**Number of children**																
2–4 (1 child, OR = 1)	1.62	1.02	2.59	0.041	1.72	1.00	2.97	0.050	0.72	0.41	1.26	0.254				
**Age of child (months)**	0.57	0.51	0.65	<0.001	0.55	0.48	0.62	<0.001	1.17	1.03	1.33	0.018	1.16	1.02	1.315	0.029
**Combined Mode and Place of delivery**																
Vaginal (Caesarean, OR = 1)	1.47	0.88	2.43	0.139					0.53	0.28	1.02	0.058				
Home	1.08	0.64	1.81	0.784					0.70	0.36	1.35	0.286				
**Type of delivery assistance**																
Non-health professional (Health professional, OR = 1)	0.76	0.50	1.14	0.183					1.53	0.91	2.58	0.108				
**Antenatal Clinic visits**																
4–7 (8+, OR = 1)	0.67	0.18	2.41	0.537					1.49	0.41	5.51	0.545				
1–3	0.51	0.16	1.64	0.257					1.20	0.34	4.26	0.781				
None	0.58	0.18	1.89	0.366					1.96	0.53	7.23	0.310				
**Postnatal check-up**																
After 2 days (0–2 days, OR = 1)	1.13	0.51	2.54	0.761	1.37	1.21	4.65	0.012	0.69	0.32	1.48	0.337				
No postnatal check-up	0.80	0.33	1.92	0.619	1.20	0.78	1.91	0.430	1.02	0.64	1.64	0.919				
**Mother’s BMI**																
18.5–24.9 (<18.5, OR = 1)	1.13	0.51	2.54	0.761					1.00	0.37	2.71	0.998				
25+	0.80	0.33	1.92	0.619					0.97	0.33	2.82	0.959				
**Exposure to Media**																
**Mother reading Newspapers**																
Yes ^#^ (Not at all, OR = 1)	0.81	0.52	1.28	0.367					1.43	0.83	2.48	0.196				
**Mother listening to the radio**																
Yes ^#^ (Not at all, OR = 1)	0.71	0.47	1.07	0.100					1.61	1.02	2.54	0.040	1.71	1.07	2.73	0.024
**Mother watching television**																
Yes ^#^ (Not at all, OR = 1)	0.83	0.57	1.21	0.336					1.15	0.74	1.77	0.531				
***Household-level factors***																
**Household Wealth Index**																
Richer (Richest, OR = 1)	1.15	0.70	1.89	0.573					0.95	0.51	1.75	0.865				
Middle	1.25	0.72	2.18	0.428					0.76	0.36	1.58	0.459				
Poorer	1.11	0.65	1.90	0.696					0.92	0.47	1.78	0.796				
Poorest	1.29	0.67	2.48	0.450					0.63	0.30	1.29	0.206				
**Decision-marking (Autonomy)**																
Some Decisions (No Decision, OR = 1)	0.67	0.45	1.01	0.055					1.43	0.89	2,28	0.137				
All Decisions	0.79	0.47	1.34	0.392					0.88	0.48	1.63	0.693				
**Residence**																
Rural (Urban, OR = 1)	1.25	0.85	1.85	0.258					1.00	0.61	1.64	0.989				
**Geographical Region without Rangpur**																
Chittagong (Barisal, OR = 1)	3.14	1.70	5.82	<0.001					0.46	0.23	0.93	0.030				
Dhaka	1.53	0.83	2.82	0.171					0.32	0.16	0.66	0.002				
Khulna	3.10	1.66	5.78	<0.001					0.53	0.25	1.10	0.089				
Rajshahi	1.42	0.78	2.59	0.254					0.41	0.20	0.84	0.015				
Sylhet	4.13	2.00	8.56	<0.001					0.58	0.26	1.29	0.178				
**Geographical Region with Rangpur**																
Chittagong (Barisal, OR = 1)	3.14	1.70	5.82	<0.001	3.19	1.59	6.41	0.001	0.46	0.23	0.93	0.030	0.45	0.22	0.92	0.029
Dhaka	1.53	0.83	2.82	0.171	1.33	0.71	2.50	0.372	0.32	0.16	0.66	0.002	0.31	0.15	0.64	0.002
Khulna	3.10	1.66	5.78	<0.001	3.62	1.80	7.29	<0.001	0.53	0.25	1.10	0.089	0.50	0.24	1.03	0.061
Rajshahi	1.42	0.78	2.59	0.254	1.34	0.69	2.62	0.389	0.41	0.20	0.84	0.015	0.35	0.18	0.80	0.011
Sylhet	4.13	1.97	8.65	<0.001	5.11	2.24	11.63	<0.001	0.58	0.26	1.30	0.184	0.57	0.25	1.28	0.173
Rangpur	6.42	2.71	15.22	<0.001	6.10	2.20	16.86	0.001	0.19	0.06	0.54	0.002	0.20	0.07	0.58	0.003

^ divorced/separated/widow, ^$^ Hinduism/Buddhism/Christianity/others; ^#^ less than once a week/at least once a week/almost every day, * values and 95%CI too wide due to small sample size.

## Data Availability

Datasets are available to at https://www.dhsprogram.com/data/.

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
