# Peer review of "Breastfeeding Practices among Adolescent Mothers and Associated Factors in Bangladesh (2004–2014)"

_nutrients, 2021, doi:10.3390/nu13020557_

Round 1
Reviewer 1 Report
The article ‘‘Breastfeeding practices among adolescents mother and associated factors in Bangladesh (2004-2014)’’ addresses an important issue regarding breastfeeding practices that may increase breastfeeding of infants at early age. The Wealth Health Organization (WHO) also recommends breast feeding for 6 months. The major concern of this study is the lack of description and information of the study population. The number of subjects is quite confusing. In page 3 it was written ‘‘A weighted total sample of 2554 adolescent mothers aged 12-19 years were included in the final analysis (2004: n=614; 2007: n=521; 2011: n=698; and 2014: n=726).’’ However, adding 2004: n=614; 2007: n=521; 2011: n=698; and 2014: n=726 will give a number of 2559.
The total number of subjects in Table 1 variates.
The number of subjects from the Bangladesh Demographic and Health Survey (BDHS) in each year should be included since bottle feeding was not collected in the 2007 BDHS. Additional these information can be included in a supplementary file.
The presentation of the results can be improved in such a way that it is easy to understand the tables and the statistical analysis used. The layout of table 2 and 3 should be revised. In table 1 and in page 4 it was written ‘‘11.7% were home deliveries, 21.5% had health professionals available at delivery, and 76.6% were delivered by caesarean section.’’ How is it possible to have only 21.5% with health professionals available at delivery when there was 76.6% delivered by caesarean section?
The number of subjects in each characteristic should also be included in the tables to understand the numbers behind each analysis.
Bottle-feeding was defined as the proportion of children aged 0-23 months who received any food or liquid including non-human milk and formula, any liquid (including breast milk) or semi-solid food from a bottle with nipple/teat. Since most of the children will receive food or formula after 12 months (1 year), there is a possibility that the results in table 3 is affected by the number of children in different age categories. Accordingly, it was shown in table 3 that children aged 18-23 months were less likely to be bottle-fed. Thus, the age of children in each characteristic should be examined.
It should be shown that children age doesn't effect the observed statistical differences between study population.
Author Response
Comments and Suggestions for Authors
The article ‘‘Breastfeeding practices among adolescents mother and associated factors in Bangladesh (2004-2014)’’ addresses an important issue regarding breastfeeding practices that may increase breastfeeding of infants at early age. The Wealth Health Organization (WHO) also recommends breast feeding for 6 months. The major concern of this study is the lack of description and information of the study population. The number of subjects is quite confusing. In page 3 it was written ‘‘A weighted total sample of 2554 adolescent mothers aged 12-19 years were included in the final analysis (2004: n=614; 2007: n=521; 2011: n=698; and 2014: n=726).’’ However, adding 2004: n=614; 2007: n=521; 2011: n=698; and 2014: n=726 will give a number of 2559.
The total number of subjects in Table 1 variates.
Response: Thanks for picking this up. It was a typo error. We wrote 698 instead of 693, and this has been edited.
The number of subjects from the Bangladesh Demographic and Health Survey (BDHS) in each year should be included since bottle feeding was not collected in the 2007 BDHS. Additional these information can be included in a supplementary file.
Response: Agreed and we have included a supplementary reflecting number of subjects for each year of the surveys
The presentation of the results can be improved in such a way that it is easy to understand the tables and the statistical analysis used. The layout of table 2 and 3 should be revised. In table 1 and in page 4 it was written ‘‘11.7% were home deliveries, 21.5% had health professionals available at delivery, and 76.6% were delivered by caesarean section.’’ How is it possible to have only 21.5% with health professionals available at delivery when there was 76.6% delivered by caesarean section?
Response: Thanks for picking this up. The percentages were swap arround. 76.6% (now 75.9) should be home delivery while 11.7% (now 12.1) were delivered by caesarean section but this has been edited in the revised manuscript.
The number of subjects in each characteristic should also be included in the tables to understand the numbers behind each analysis.
Response: Agreed and we have included the text below as a footnote in table 1.
“Total count 2554 unless otherwise given in brackets”
Bottle-feeding was defined as the proportion of children aged 0-23 months who received any food or liquid including non-human milk and formula, any liquid (including breast milk) or semi-solid food from a bottle with nipple/teat. Since most of the children will receive food or formula after 12 months (1 year), there is a possibility that the results in table 3 is affected by the number of children in different age categories. Accordingly, it was shown in table 3 that children aged 18-23 months were less likely to be bottle-fed. Thus, the age of children in each characteristic should be examined.
Response: Agreed and text now edited as requested by the reviewer.
It should be shown that children age doesn't effect the observed statistical differences between study population.
Response: Agreed and we have added the text below in the discussion section.
the main effect of children age in category did not differ statistically (Adjusted Wald χ2 = 2.22, P= 0.085

Reviewer 2 Report
Emwinyore Agho et al article reported observational data related to breastfeeding practices in Bangladesh. All data related this field is important to optimize what factors are associated to increase not only the quality of the breast milk but also the maternal health during lactation. However, this data is provided from survey and national repository and, as all studies, could get some weakness which, kindly I want to comments with the authors.
Mayor comments:
- Introduction section: In general, I would suggest to summarize the text, sometimes the paragraph distract from the article focus (i.e. “The adolescent period (aged between 10 and 19 years) is a time when young people are becoming more independent, develop social skills and forge new lifelong relationships. This period also present challenges compared to adulthood, as adolescents are more likely to experience drugs and alcohol, mental health and/or sexual health issues such as adolescent pregnancy/motherhood [1].”). I kindly suggest to remove it. In addition, the prevalence of new maternity in adolescent around the country or particularly in LMIC should be reported.
- Material and Methods section: these section has many variables collected. I suggest if the authors could re-organize the variable and defined one by one. Some of them are confusing (i.e. what BDHS variables were collected exactly? the hv271 variable is not clear how was scored, what does the index mean?) in addition, other variables such as: gestational age, Apgar, birth weight, parity, gravity, miscarriage, etc… were collected?
- The statistical analysis is difficult to follow. Kindly suggest to re-organize it.
- Result section:
- I do not know if the table 1 report essential information related to manuscript proposal. In addition, What the authors mean with primary/secondary education? Why the mother age was categorized (same for child age, maternal BMI…)? What difference there was between marital status? The birth order is the parity? In addition, to simplify the tables, for the dichotomy variables the authors just need to inform of one of two categories.
- The table 2 correspondent with the description of the BF practices, I kindly suggest re-organized the table (i.e.: column as % and (n), just one line per row and what 95% CI was calculated?) the data related to EBF below 6 months is interesting and also de median for BF practice (in month). Similarly, the figure 1 looks interesting but the understandability is rare. I kindly suggest, the same prevalence but by lines.
- In the table 3 and 4, I kindly suggest to remove the reference in every predictive variable for the models, because always will be OR=1. In addition, it would be useful if the footnote reports what variables were considered to adjust the models. in other hand, the variable “Decision-marking (Autonomy)” need to be clarified because there is relationship with the outcomes of the study.
- Discussion section:
- It is not clear why Some Decisions could be a protective factor while All Decisions would be a risk factor in the adjusted models for delayed initiation of BM. In addition, have make sense that in general, more maternal age, increase educational level, more clinical visit (medical support: related with no-home delivery) could be related to better BM practices such as (no-delayed BM initiation or non-bottle-feedings). However, for the EBF and predominant BM models, the factors (child age and make decision) have opposite trends, what could be the rationality according to the authors knowledge?
- The prevalence of EBF in this cohort is concordance with similar countries? There was improvement along the years? That could be related to educational level, economy income, family support, etc or it is more property of law-issue in each country? I suggest consult PMID: 33007816.
- Conclusion section: this section should be re-written to give objective information or even strongly recommendation according to their results in Bangladesh BF practice for adolescent mothers/clinical care staff.
Minor comments:
- EBF acronyms need to be defined at the first time.
- The article has many acronyms; it would be useful if the authors could write an acronyms list.
Author Response
Mayor comments:
- Introduction section: In general, I would suggest to summarize the text, sometimes the paragraph distract from the article focus (i.e. “The adolescent period (aged between 10 and 19 years) is a time when young people are becoming more independent, develop social skills and forge new lifelong relationships. This period also present challenges compared to adulthood, as adolescents are more likely to experience drugs and alcohol, mental health and/or sexual health issues such as adolescent pregnancy/motherhood [1].”). I kindly suggest to remove it. In addition, the prevalence of new maternity in adolescent around the country or particularly in LMIC should be reported.
Response:Text now edited as requested by the reviewer.
- Material and Methods section: these section has many variables collected. I suggest if the authors could re-organize the variable and defined one by one. Some of them are confusing (i.e. what BDHS variables were collected exactly? the hv271 variable is not clear how was scored, what does the index mean?) in addition, other variables such as: gestational age, Apgar, birth weight, parity, gravity, miscarriage, etc… were collected?
Response: Relevant sections of the Methods have revised as requested by the reviewer. The the hv271 variable and estimation method has also been clarified in the revised manuscript. As we have discussed in the strengths and limitations section, this study was based on on a secondary analysis of the Bangladesh Demographic and Health Survey (BDHS) data. This means that our analyses and/or results would be limited to available variables. The BDHS collects data on fertility, reproductive health, maternal and child health, mortality, nutrition and self-reported health behaviours among adults. The survey, however, did not collect information on some maternal and infant health outcomes (such as gestational age, Apgar, birth weight, parity, gravity and miscarriage). We agree with the reviewer that this additional information may have been useful in the study, and thus, have incorporated this information in the limitation section of the revised manuscript.
- The statistical analysis is difficult to follow. Kindly suggest to re-organize it.
Response: For clearity, we have re-organised the statistical section.
- Result section:
- I do not know if the table 1 report essential information related to manuscript proposal. In addition, What the authors mean with primary/secondary education? Why the mother age was categorized (same for child age, maternal BMI…)? What difference there was between marital status? The birth order is the parity? In addition, to simplify the tables, for the dichotomy variables the authors just need to inform of one of two categories.
Response: We have included a supplementary table that detailed the “Definition and categorisation of potential variables used in the study”
- The table 2 correspondent with the description of the BF practices, I kindly suggest re-organized the table (i.e.: column as % and (n), just one line per row and what 95% CI was calculated?) the data related to EBF below 6 months is interesting and also de median for BF practice (in month).
Response: Agreed and the table has been re-orgainsed - one line per row. We have edited the 95% CI to reflect the fact that 95% CI was calucated for rates
Similarly, the figure 1 looks interesting but the understandability is rare. I kindly suggest, the same prevalence but by lines
Response: Figure 1 is the conventional method for reporting breasting practices in the DHS and UNICEF by age – please see page 160 in this link below. The reason for producing this graph is that from this specific graph, a layperson could easily determine the approximate median age of EBF or anyBF by drawing a straight line across 50% on the vertical axis.
https://dhsprogram.com/pubs/pdf/FR311/FR311.pdf.
- In the table 3 and 4, I kindly suggest to remove the reference in every predictive variable for the models, because always will be OR=1. In addition, it would be useful if the footnote reports what variables were considered to adjust the models. in other hand, the variable
Response: Tables 3 and 4 has been edited
- “Decision-marking (Autonomy)” need to be clarified because there is relationship with the outcomes of the study.
Response: We have modified decision making variable by reducing the variables from 5 to 3 including recalculating the women's decision-making autonomy and re-analysising EIBF, EBF and PBF. In addition, we have included, supplementary table that detailed the “Definition and categorisation of potential variables used in the study”
- Discussion section:
- It is not clear why Some Decisions could be a protective factor while All Decisions would be a risk factor in the adjusted models for delayed initiation of BM. In addition, have make sense that in general, more maternal age, increase educational level, more clinical visit (medical support: related with no-home delivery) could be related to better BM practices such as (no-delayed BM initiation or non-bottle-feedings).
Response: Some Decisions no longer significant after modification of definiation and analysis
However, for the EBF and predominant BM models, the factors (child age and make decision) have opposite trends, what could be the rationality according to the authors knowledge?
Response: EBF is protective of mortality and morbidity while, PBF is a risk factor for mortality and morbidity from past studies. However, the text is now edited as requested by the reviewer.
- The prevalence of EBF in this cohort is concordance with similar countries? There was improvement along the years? That could be related to educational level, economy income, family support, etc or it is more property of law-issue in each country? I suggest consult PMID: 33007816.
Response: : Text now edited as requested by the reviewer
- Conclusion section: this section should be re-written to give objective information or even strongly recommendation according to their results in Bangladesh BF practice for adolescent mothers/clinical care staff.
Response: Conclusion has been modified to include health workers.
Minor comments:
- EBF acronyms need to be defined at the first time.
Response: Done
- The article has many acronyms; it would be useful if the authors could write an acronyms list.
Response: We have checked journal procedure and it indicated that “Abbreviations should be defined in parentheses the first time they appear in the abstract, main text, and in figure or table captions and used consistently thereafter”.

Round 2
Reviewer 2 Report
Thank you for this second version which has improved some of the points considered.
I am still in doubt whether figure 1 is fully understandable. As reported by the authors on page 160 of the pdf (year 2014), a similar graph is reported but added to previous table 11.3 that complement the information. Similarly, the graphs of % tendencies are reported both in lines as well as bars (page 156).
On the other hand, although the journal's author guidelines recommend defining abbreviations in the text, it is not incompatible with a list of abbreviations. Please consult https://doi.org/10.3390/nu13010205
Author Response
Thanks for the quick feedback.
I am still in doubt whether figure 1 is fully understandable. As reported by the authors on page 160 of the pdf (year 2014), a similar graph is reported but added to previous table 11.3 that complement the information. Similarly, the graphs of % tendencies are reported both in lines as well as bars (page 156).
Response: Agreed and for clarity, we have included a supplementary table 4 which shows the distribution of breastfeeding by child age in months.
On the other hand, although the journal's author guidelines recommend defining abbreviations in the text, it is not incompatible with a list of abbreviations. Please consult
https://doi.org/10.3390/nu13010205
Response: thanks for the useful link and list of abbreviations now included before references.